# The ALFF Alterations of Spontaneous Pelvic Pain in the Patients of Chronic Prostatitis/Chronic Pelvic Pain Syndrome Evaluated by fMRI

**DOI:** 10.3390/brainsci12101344

**Published:** 2022-10-04

**Authors:** Shengyang Ge, Qingfeng Hu, Guowei Xia, Yifan Tan, Yijun Guo, Chuanyu Sun

**Affiliations:** 1Department of Urology, Huashan Hospital, Fudan University, 12 Urumqi Middle Rd., Shanghai 200040, China; 2Department of Urology, Jing’an District Central Hospital, Fudan University, 259 Xikang Rd., Shanghai 200040, China

**Keywords:** chronic prostatitis/chronic pelvic pain syndrome, functional magnetic resonance imaging, amplitude of low-frequency fluctuation, fractional amplitude of low-frequency fluctuation

## Abstract

Chronic prostatitis/chronic pelvic pain syndrome (CP/CPPS) is a challenging entity with complicated symptoms for treatment in the male crowd. Accumulating evidence revealed the dysfunction in the central system should be a critical factor for the pathogenesis and development in the CP/CPPS. Therefore, we recruited 20 patients of CP/CPPS and 20 healthy male volunteers, aged 20 to 50 years. Through resting-state functional magnetic resonance imaging (fMRI), we analyzed the mean amplitude of low-frequency fluctuations (mALFF) and the mean fractional amplitude of low-frequency fluctuations (mfALFF) to reflect the spontaneous abnormal activated regions in the brains of CP/CPPS patients. Compared to the healthy controls, the group with CP/CPPS had significantly increased mALFF values in the thalamus and augmented fALFF values in the inferior parietal lobule and cingulate gyrus. Significant positive correlations were observed in the extracted mALFF values in the midbrain periaqueductal gray matter (PAG) and the pain intensity (*r* = 0.2712, *p* = 0.0019), mALFF values in the thalamus and the scores of Hospital Anxiety and Depression Scale (HADS) anxiety subscale (*r* = 0.08477, *p* = 0.0461), and mfALFF values in the superior frontal gyrus (SFG) and the scores of the HADS anxiety subscale (*r* = 0.07102, *p* = 0.0282). Therefore, we delineated the clinical alterations in patients of CP/CPPS that might be attributed to the functional abnormality of the thalamus, inferior parietal lobule, and cingulate gyrus. Among these regions, the PAG, thalamus, and SFG may further play an important role in the pathogenesis, with their regulating effect on pain or emotion.

## 1. Introduction

Chronic prostatitis/chronic pelvic pain syndrome (CP/CPPS) is one of the most common disorders in the daily clinic of urology, which occupies approximately 90–95% types of chronic prostatitis [1]. On the basis of the official definition from the National Institutes of Health (NIH), the features of typical symptoms of CP/CPPS should contain chronic pain symptoms in the pelvic and/or genital regions lasting more than 3 months during the past 6 months without any other possibly detectible causes, usually accompanying other symptoms, such as lower urinary tract symptoms and/or sexual dysfunction [2]. CP/CPPS is a kind of chronic pain disease, so the patients of CP/CPPS also manifest various psychiatric and/or psychosocial symptoms [3]. Due to the chronic pain and prevalent symptoms caused by CP/CPPS, the life quality of these male patients could be significantly influenced [4].

Although in the past decades many different etiologies and mechanisms of pathogenesis of CP/CPPS have been proposed, the pathogenesis of CP/CPPS remains unclear [5]. A mass of evidence revealed several aspects of alterations participating in the pathogenesis and development of CP/CPPS, which includes neuroendocrine, infective, inflammatory, autoimmunity-related, and psychosocial factors [6,7,8,9]. Presently, the pharmacological or non-pharmacological approaches of treating CP/CPPS are merely symptomatic, although there are several treatments ever developed which contain drugs, such as non-steroidal anti-inflammatory drugs (NSAIDs), α-blockers, antibiotics, and phytotherapy, and alternative therapies, such as acupuncture and moxibustion [10,11]. Attributing to the individual heterogeneity and complicated mechanisms of CP/CPPS, no monotherapy could bring reliable and persistent profits for the patients [12]. Therefore, the further investigation for the etiology and pathogenesis of CP/CPPS could bring the improvement of diagnosis and treatment.

Because CP/CPPS belongs to a kind of chronic pain disorder, central sensitization mechanisms, such as neuromodulation and reorganization in the brain, have been proven to concern the pathophysiological progresses [13]. Resting-state functional magnetic resonance imaging (fMRI) with higher magnetic field strengths is a noninvasive and accurate technique that could reflect brain changes consisting of temporally synchronous, spatially distributed, and spontaneous blood-oxygen-level-dependent (BOLD) signal fluctuations at rest condition [14]. Through fMRI, several functional and structural alterations related with different brain properties in the neural nervous system have been qualified in patients with chronic pain disorders, such as chronic musculoskeletal pain, CP/CPPS, and chronic low back pain [15,16]. In patients of CP/CPPS, specific patterns of an impaired pain modulatory system, abnormal activated regions, and anatomical brain reorganization were reported [17,18].

The spontaneous low-frequency (typically 0.01–0.08 Hz) oscillations (LFOs) of the human brain are thought to reflect the changes in spontaneous neuronal and physiological activities to a certain extent [19]. The amplitude of low-frequency fluctuations (ALFF) and the fractional amplitude of low-frequency fluctuations (fALFF) were applied to evaluate the spontaneous brain activities [20]. As a reliable measuring method for observation, resting-state fMRI could imitate altered brain areas by evaluating a synchronous ultra-slow frequency oscillation according to the theory of hemodynamics. For a time series, the ALFF measures the amplitude of time-series fluctuations at each voxel within the LFOs [21]. The fALFF is defined as the total power within the LFOs divided by the total power in the entire detectable frequency range, which is determined by the sampling rate and duration. As a normalized index of the ALFF, the fALFF can provide a more specific measure of the low-frequency oscillatory phenomena [22]. Because the fALFF approach could selectively suppress artifacts from non-specific brain areas, it could enhance signals from the cortical regions associated with brain activity, making use of the distinct characteristics of their signals in the frequency domain [23].

Our previous work revealed that altered functional connectivities in the patients with CP/CPPS might play a vital role in the pathogenesis and development of CP/CPPS by analyzing the integrity of the default mode network (DMN) [24]. We postulated that the main cause of chronic pain in CP/CPPS might ascribe to the long-term hemodynamic changes and neural plasticity in the central nervous system which could be detected in brain region levels by resting-state fMRI. Therefore, we recruited 20 CP/CPPS patients with spontaneous pelvic pain, together with 20 age-matched healthy volunteers, to discover the possible functional alterations of the brain in the resting state and elucidate the potential mechanism of the pathogenesis and development of CP/CPPS.

## 2. Materials and Methods

### 2.1. Characteristics of Participants

We actually recruited a cohort of 20 male right-handed patients with spontaneous pelvic pain caused by chronic prostatitis/chronic pelvic pain syndrome and 20 healthy right-handed, age- and gender-matched controls (Table 1). The patients of CP/CPPS were straightly recruited by oral inquiry and fully informed of the project in the daily clinic of Huashan Hospital, Shanghai, China, whilst the group of healthy control was recruited by poster advertising. This study was approved and supervised by the Ethics Committee of Huashan Hospital, Fudan University (Ethics Approval No. 2021-772.) and the Ethics Committee of Jing’an District Central Hospital, Fudan University (Ethics Approval No. 2020-05). All enrolled participants signed the written informed consents before data acquisition. Chief complaint, physical examinations, routine urine, standard microbiological cultures of urine, and transrectal ultrasonography (TRUS) were used for the diagnosis of CP/CPPS and the exclusion of acute or chronic bacterial prostatitis, benign prostate hyperplasia, prostate cancer, and other pelvic diseases. The participants were excluded if they admitted to having any other chronic pain disorders, history of malignant tumors, and chronic diseases that might lead to peripheral nerve injury, such as diabetes mellitus and hypertension. All the patients declared they did not take any medications or alternative treatment for CP/CPPS before or stopped taking the therapies to treat CP/CPPS for more than a month, and they denied the other discomforts at any other body parts. After obtaining the written informed consent, all of the participants were requested to finish two clinic scales, one National Institutes of Health Chronic Prostatitis Symptom Index (NIH-CPSI) scale and one Hospital Anxiety and Depression Scale (HADS). All the scales were deposited for safekeeping, the privacy of the participants was not compromised, and all research was performed in accordance with relevant guidelines and regulations. According to the previous study, 15 participants in each group are demanded for the minimum size in neuroimaging studies [25]. Then, we calculated the sample size according to the remaining research funds.

### 2.2. The Adapted Clinical Scales

*NIH-CPSI scale.* The NIH-CPSI was an objective assessment tool and outcome measurement for prostatitis-like symptoms in the management of men with CP/CPPS since 1999 [26]. The main components of NIH-CPSI encompass pain with 4 items focused on location, severity, and frequency; urinary function, embracing one irritative item and one obstructive item; and quality of life impact, with 3 items about the effect of symptoms on daily activities [26]. The NIH-CPSI is commonly used in clinical trials, along with in the evidence-based evaluation of treatment effects [27]. The higher scores of NIH-CPSI scale revealed the worse prostatitis-like symptoms the patients were suffering. 

*NRS.* There is a built-in Numeric Rating Scale (NRS) in the NIH-CPSI scale. To assess the intensity and severity of spontaneous pain degree, the NRS is a segmented numeric version of the visual analog scale (VAS) by selecting a whole number (0–10 integers) to describe the pain degree [28]. We listed the scores of NRS separately to reflect the spontaneous pain intensity in the cohort.

*HADS.* In the patients of CP/CPPS, the emotional symptoms, such as anxiety and depression, are always accompanying; we evaluated the mental issues in the participants by the HADS questionnaire. It was because HADS was the most useful and widely used method in both the literature and clinical health care [29]. The scores of each subscale in HADS would not be meaningful unless the score was more than 7. Then, the higher scores of HADS suggested the worse psychological condition of the participants.

### 2.3. Resting-State fMRI Data Acquisition

fMRI data were obtained from 20 patients with chronic prostatitis/chronic pelvic pain syndrome and 20 healthy volunteers, and participants rated spontaneous pain inside the scanner. The resting-state fMRI data were obtained by using a 3.0 T GE MR 750 MRI scanner with an eight-channel phase array head coil at the Jingan District Centre Hospital, Shanghai. Whole resting-state fMRI data were acquired using a gradient-recalled echo-planar imaging pulse sequence (repetition time (TR)/echo time (TE) = 2000/30 ms; FA = 90°; acquisition matrix = 64 × 64; field of view (FOV) = 24 × 24 cm^2^; slice thickness = 4 mm; no gap; 38 slices and total 210 time points). The high-resolution T1-weighted magnetic resonance images were collected by a three-dimensional fast spoiled gradient-echo dual-echo sequence (TR = 8.2 ms; TE = 3.2 ms; matrix = 256 × 256; FOV = 24 × 24 cm^2^; slice thickness = 1 mm; no gap and 156 slices). To classify individual participants, it was feasible for the authors to approach the personal information during or after data collection.

### 2.4. Data Preprocessing Analysis

All the DICOM files in format were converted to NIFTI files by MRIconvert (http://lcni.uoregon.edu/jolinda/MRIConvert/ (accessed on 1 May 2022)). The resting-state fMRI data were also preprocessed using SPM12 (https://www.fil.ion.ucl.ac.uk/spm/ (accessed on 12 June 2022)) and RESTPlus V1.22 (http://www.restfmri.net (accessed on 12 June 2022)). To avoid statistic errors, all the scans were well-examined before preprocessed. Unfortunately, 2 sets of data in patient group were abandoned because of image distortion. The preprocessing steps included: (1) discarding the first 10 timepoints for reaching a steady-state magnetization and allowing all the participants to adapt to scanning noise, and the remaining 190 time points of image were processed in our following study; (2) slice timing correction, slice order could be Matlab formula: [1:2:43 2:2:42], for eliminating the difference in the time in differentiated fMRI signal levels, and the signals were calibrated as if they were obtained at the same time point; (3) head motion correction; (4) normalization, transforming the brain images to reduce the variability between individuals and allowing meaningful group analyses by using T1 image unified segmentation, with bounding box [−90,−126,−72;90,90,108] and isotropic voxel size [3,3,3]; (5) spatial smoothing, by the convolution of the three-dimensional image with a three-dimensional Gaussian kernel with a full width at half maximum (FWHM) of 6 mm; (6) removing the linear trend of the time series caused by warming of the scanner or adaptation of the participants, with the time accumulation; (7) nuisance covariables regression, including 6 head motion parameters, the cerebrospinal flow signals, and white matter signals. The mean value of the time series of each voxel was not added back in this step. After preprocessing in RestPlus, linear trend was removed.

### 2.5. Analysis of ALFF and fALFF

ALFF and fALFF analyses were performed using the RESTplus V1.22 after data preprocessing. The time series for each voxel was transformed to the frequency domain and the power spectrum was then obtained. The square root was calculated at each frequency of the power spectrum and the averaged square root was obtained across 0.01–0.08 Hz at each voxel. This averaged square root was taken as the ALFF. Because the scale of BOLD signal can affect the ALFF value, the ALFF value was further divided by the global mean ALFF (mALFF) value to standardize the data. The procedure of data analysis of fALFF was similar to ALFF. After taken as the ALFF, the amplitude of the current band (0.01–0.08 Hz) divided by the amplitude of the full band (0–0.25 Hz), which made the fALFF and mean fALFF (mfALFF).

### 2.6. Statistics Analysis

A two-sample t-test was conducted based on the measured data of clinic scales or other basic information (ages, duration of illness, the scores of NIH-CPSI, NRS, HADS, and their subscales) by SPSS 17.0, and results were scheduled as $\bar{X}$ ± S (Table 1). A two-sample t-test was conducted after the analysis of ALFF and fALFF. False Discovery Rates (FDR), as multiple comparison correction, corrected the values of abnormal brain regions (voxel *p* < 0.01, cluster *p* < 0.01). The statistical images were superimposed on the standard spatial template for presentation. Active clusters larger than 100 voxels were illustrated in the calibrated standard brain map with pseudo-color, and their Montreal Neurological Institute (MNI)coordinates, positions, and voxel sizes of peak intensity were listed in a table reported by XjView95 software (http://www.alivelearn.net/xjview (accessed on 28 June 2022)); significant difference was set at *p* < 0.05. The abnormal brain regions were respectively set as the regions of interest (ROIs). Linear regression analysis was conducted with SPSS 11.0 as dependent variables to the scores of clinical scales and the relative intensity values extracted from ROIs. The significant difference was set at *p* < 0.05. The outcomes of linear regression analysis were displayed by GraphPad 8.0.

## 3. Results

### 3.1. Features of Observational Cohorts

As demonstrated in Table 1, the recruited cohort of the CP/CPPS patients and the healthy control were the same gender, right-handed, and age-matched (*p* = 0.1497). To elude the potential impact from senile brain atrophy, the age ranges of the participants were set from 20 to 50 years old. The baseline for the duration of spontaneous pelvic pain was about 20 months, which was consistent with the foundational description of CP/CPPS.

The mean scores of the total NIH-CPSI were approximately 28, which revealed that the participated patients with CP/CPPS endured a severe disease condition when the scores were over 18. Because the counted scores of the NRS in the cohort with CP/CPPS were 3.50 ± 2.35, the pain degrees of spontaneous pelvic pain were between mild and moderate states. Moreover, the NRS also assisted us to eliminate the potential chronic pain patients in the confirmation of healthy volunteers. The scores of the HADS subscale will be valuable if the corresponding score is over 7, so the patient cohort presented the psychological feature of anxiety with the mean scores of the HADS anxiety subscale (9.44 ± 5.54). The relatively low mean scores of the HADS depression subscale may be because of a short disease course and outpatient bias. Hereafter, subsequent attention will be paid to the psychological situation of anxiety rather than depression in the CP/CPPS participants.

### 3.2. The Abnormal Activated Brain Regions by mALFF Analysis

As the altered clusters were illustrated in Figure 1 and Table 2, we found the brain region of the thalamus was the majority of the abnormal activated position. Moreover, the most focal positive activated area of the thalamus was perceived in the dorsomedial region. Except for the area of the thalamus, this cluster also covered the parts of the putamen, midbrain, parahippocampal gyrus, right amygdala, right pallidum, etc. No negative activated brain region was discovered in this cohort.

### 3.3. The Abnormal Activated Brain Regions by mfALFF Analysis

As demonstrated in Table 2 and Figure 2, several clusters of highly activated brain regions were also detected by the mfALFF analysis. The major altered regions mainly distributed in the inferior parietal lobule and cingulate gyrus. In the areas of the inferior parietal lobule, the altered brain regions got involved in the precentral gyrus, postcentral gyrus, anterior cingulate, superior frontal gyrus (SFG), and medial frontal gyrus. In the areas of the cingulate gyrus, the altered brain regions included the SFG, frontal superior gyrus, left frontal superior medial gyrus, and medial frontal gyrus. There was no negative activated brain region in this analysis.

### 3.4. The Correlation Analysis of the Extracted Values in the Abnormal Brain Regions and Clinical Scale Scores

The values of the mALFF and mfALFF were extracted in the correspondingly picked clusters. As exhibited in Figure 3a, the values of the abnormal mALFF in the picked region of the midbrain periaqueductal gray matter (PAG) showed a positive correlation with the scores of the NRS (*r* = 0.2712, *p* = 0.0019), which revealed the abnormality of the PAG affected the pain degrees of the CP/CPPS patients with spontaneous pelvic pain. As presented in Figure 3b, the values of the abnormal mALFF in the cluster of the thalamus displayed a positive correlation with the scores of the HADS anxiety subscale (*r* = 0.08477, *p* = 0.0461), which implied that the emotional changes in the cohort of the CP/CPPS patients might attribute to the impact of the altered function in the thalamus. Similarly, as demonstrated in Figure 3c, the values of the abnormal mfALFF in the cluster of the SFG revealed a positive correlation with the scores of the HADS anxiety subscale (*r* = 0.07102, *p* = 0.0461), which hinted the anxiety symptoms in the participated patient group would also refer to the alterations in the identified SFG.

## 4. Discussion

CP/CPPS are the most common types of urological diseases in the male population, which generates several challenging issues, such as an obscure pathogenesis, a lack of effective therapies, high recurrence rates after treating, emotional symptoms, the low life qualities of patients, and a heavy social burden [30]. In recent years, CP/CPPS was discovered to be associated with the alterations in brain activity and excitability, which revealed that brain functional reorganization and neuroplasticity should play a vital role in the pain modulation of CP/CPPS [17,31].

As illustrated in Figure 1, the cluster of the PAG in the midbrain was also identified. Although the extracted values of this ROI were not correlated with the emotional scales, the correlation between the scores of the NRS and the values of the mALFF displayed the participating role of the dysfunctional PAG in the pain perception of CP/CPPS patients. The PAG is an essential hub in the pain processing network involved in descending antinociceptive functions and ascending connections with cortices [32]. In a molecule, the PAG could provide an intermediate pain control effect through its antinociceptive endogenous opioid and non-opioid pathways [33]. In the other chronic pain diseases, such as chronic low back pain, the region of the PAG was proven to be activated by noxious stimuli and related to the pathological pain states [34]. Hence, the influence of the abnormal function in the PAG might cause mild-to-moderate alterations in the clinical pain scales among the enrolled patients with CP/CPPS.

The clusters in the thalamus were the majority in the altered brain regions in the analysis of the mALFF. As a part of the limbic system, the thalamus was recognized as an important node to modulate the ascending nociceptive information in the chronicity of pain [35]. Several studies in patients and animal models have also proved that the thalamus was positively activated during chronic pain [36,37]. It was reported that the circuit of the rostral anterior cingulate cortex (rACC)-thalamus was associated with the process of the affective components but was not involved in the changes in pain intensity [38]. The increased activities of the thalamic cortex could also cause the increased activities of the insula and pain constant perception [39]. The thalamus is closely connected with the amygdala, which plays a vital role in producing fear, anxiety, and other emotional responses, such as guilt [40]. The augmented mALFF value of the right amygdala was also detected, which disclosed the potential functional relationship between the thalamus and the right amygdala in the anxiety disorder of CP/CPPS. Therefore, our outcomes revealed the local low-frequency oscillations of the thalamic regions might be responsible for the changes in the anxiety-like disorder for the patients with CP/CPPS, but the structural circuits of the thalamus and other brain components remain obscure, which may develop the potential target in the future.

Compared to the healthy control, this cluster also covered parts of the putamen, parahippocampal gyrus, right pallidum, etc. These pain-related regions also got involved in the changes in emotion and pain intensity. Likewise, the putamen participates in the sensory-discriminative parts of pain, and the increased function of the putamen may explain the clinical pain and motor impairment in patients with complex regional pain syndrome [41]. The parahippocampal gyrus is a compound of the limbic system, and an extensive activation of the limbic system aroused by mood stimuli during a migraine attack was found in migraine patients [42]. 

In Figure 2, the clusters in the inferior parietal lobule and cingulate gyrus were the major changed brain regions. The activation of the anterior cingulate cortex was found in chronic pain contributing to chronic pain states [43]. In the vast dysfunctional areas of both the cingulate gyrus and inferior parietal lobule, it revealed that the abnormal activation of brain regions in the somatosensory cortices, emotional arousal cortices, and executive cognitive networks should be participating in the pathogenesis of CP/CPPS [44]. 

As a crucial number of emotional arousal networks, the superior frontal gyrus is crucial for self-awareness and sensory and emotional processes [45]. Particularly, the SFG had been found to play a vital role in bipolar disorder with increased fALFF values because the superior frontal gyrus might be a key region related with perceived stress and further cognitive and emotional control [46]. Nevertheless, different from our outcomes that the abnormally ascending fALFF values of the SFG correlated with the scores of clinical anxiety scales, patients of major depressive disorder with gastrointestinal symptoms presented more severe depressive symptoms with an increased fALFF in the right SFG [47]. Hence, we hypothesized the dysfunction of the SFG might imply the potential tendency of translating the anxiety into depression in the observed patients of CP/CPPS. Moreover, the SFG could be regarded as a potential biomarker of depressive symptoms aroused by the indirect effect of perceived stress [48]. Further, we would follow up the enrolled cohort to remind them to concentrate on their emotional states. With more detailed clinical scales in both perceived stress and self-persuasion, monitoring the spontaneous activity of the SFG may assist in discovering the process of gradual change from anxiety to depression in the patient cohort with CP/CPPS.

We should mention there are some limitations in this study. Firstly, the participants we selected in the patients group revealed anxious symptoms, which may not cover all the emotional morbid states in the CP/CPPS. Secondly, the sample size was relatively small. The small sample size may have limited the statistical power. Then, as observational research, we did not design any intervention to assess the relationship between the abnormal brain regions and clinical prognosis. Finally, we should design an extensive study for investigating the central mechanisms of CP/CPPS for further research.

## 5. Conclusions

In conclusion, the ALFF and fALFF analysis of the resting-state fMRI revealed an abnormal pain perception system and emotional arousal network may be responsible for the symptoms of pain and anxiety in CP/CPPS. These finding could be potential parameters for assessing the severity of CP/CPPS and prospective biomarkers for further clinical observation before and after intervention.

## Figures and Tables

**Figure 1 brainsci-12-01344-f001:**
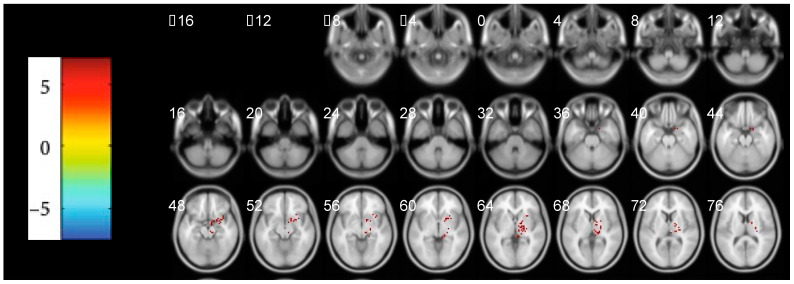
Significant alterations of mALFF between the PT and HC groups (*p* < 0.05, FDR Correction, *p* < 0.05) in the transverse position. The abnormal regions of increased mALFF values were marked in red colors. The darkness of red color implied the intensity of mALFF. mALFF: mean amplitude of low-frequency fluctuation, FDR: False Discovery Rates, PT: patient group, HC: healthy control.

**Figure 2 brainsci-12-01344-f002:**
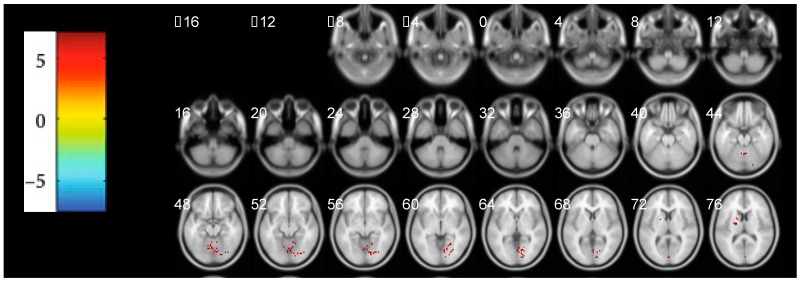
Significant alterations in mfALFF between the PT and HC groups (*p* < 0.05, FDR Correction, *p* < 0.05) in the transverse position. Regions of red colors represented these areas had increased high values of mfALFF. Likewise, the darker the red color, the higher intensities of mfALFF. mfALFF: mean fractional amplitude of low-frequency fluctuation, FDR: False Discovery Rates, PT: patient group, HC: healthy control.

**Figure 3 brainsci-12-01344-f003:**
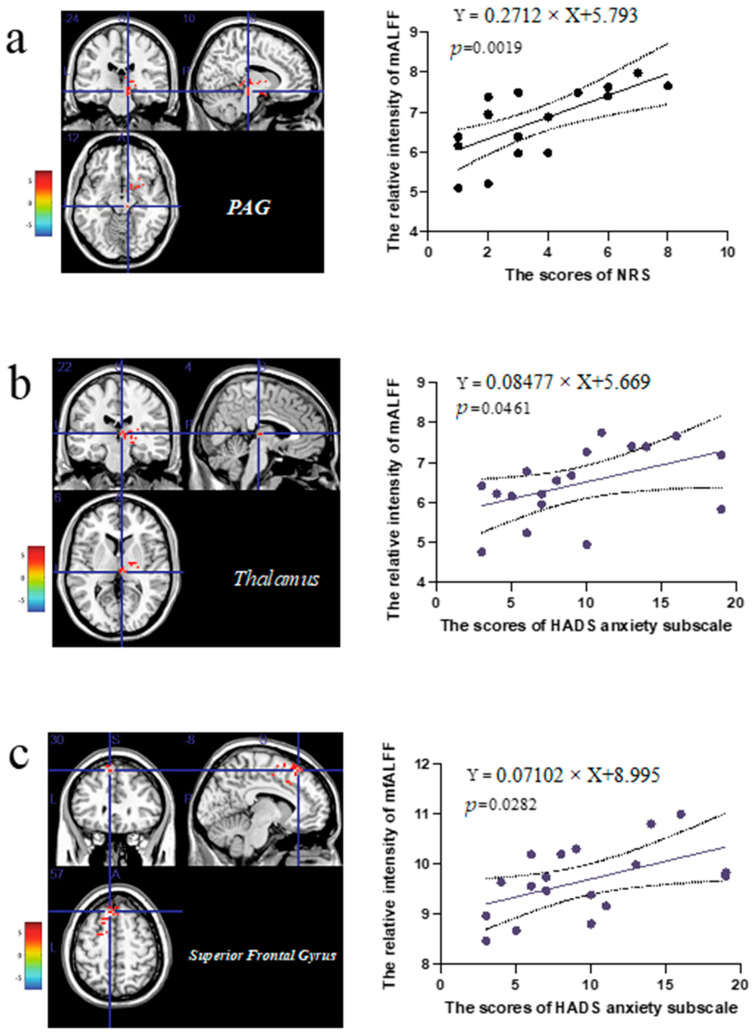
The Outcomes of Correlation Analysis among ROIs and Clinical Scale Scores. (**a**) The correlation between the scores of NRS and mALFF values in PAG (*r* = 0.2712, *p* = 0.0019, R^2^ = 0.4620). (**b**) The correlation between the scores of HADS anxiety subscale and mALFF values in thalamus (*r* = 0.08477, *p* = 0.0461, R^2^ = 0.2261). (**c**) The correlation between the scores of HADS anxiety subscale and mfALFF values in superior frontal gyrus (*r* = 0.07102, *p* = 0.0282, R^2^ = 0.2667). There was no correlation between the relative values of mALFF and mALFF in the other ROIs and the scores of clinical scales. ROIs: regions of interest, NRS: Numeric Rating Scale, mALFF: mean amplitude of low-frequency fluctuation, PAG: periaqueductal gray matter, HADS: Hospital Anxiety and Depression Scale, mfALFF: mean fractional amplitude of low-frequency fluctuation.

**Table 1 brainsci-12-01344-t001:** The Characteristics of Participants.

	Patients of Chronic Prostatitis/Chronic Pelvic Pain (*n* = 18)	Healthy Control (*n* = 20)	*p* Value
Gender	Male	Male	-
Age/years	34.28± 8.61	39.67 ± 13.32	0.1497
Duration of spontaneous pelvic pain /months	19.56 ± 5.38	0	<0.001
**Total National Institutes of Health Chronic Prostatitis Symptom Index (NIH-CPSI) score** **(item1 + 2 + 3 + 4 + 5 + 6 + 7 + 8 + 9)**	**27.78 ± 11.60**	**0**	**<0.001**
Pain and discomfort (1 + 2 + 3 + 4)	13.27 ± 8.29	0	<0.001
NRS (4)	3.50 ± 2.35	0	<0.001
Lower urinary tract symptoms (5 + 6)	6.61 ± 3.05	0	<0.001
Impact on quality of life (7 + 8 + 9)	8.44 ± 3.58	0	<0.001
Severity of symptoms (1 + 2 + 3 + 4 + 5 + 6)	20.33 ± 10.39	0	<0.001
**Hospital Anxiety and Depression Scale (HADS)**	**15.32 ± 7.99**	**0**	**<0.001**
HADS (anxiety)	9.44 ± 5.54	0	<0.001
HADS (depression)	6.17 ± 3.60	0	<0.001

**Table 2 brainsci-12-01344-t002:** The Detailed Cluster Locations of the Abnormal Brain Regions.

	Region Label	Cluster/Voxels	Peak *t*-Value	Montreal Neurological Institute (MNI) Coordinates		
				x	y	z
Positive	Thalamus	158	9.730	6	−21	9
	Inferior Parietal Lobule	220	10.612	−45	−27	24
	Cingulate Gyrus	110	7.665	−6	−15	45

## Data Availability

The datasets used and analyzed in the current study are available from the corresponding author on reasonable request.

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
