# Peer review of "The ALFF Alterations of Spontaneous Pelvic Pain in the Patients of Chronic Prostatitis/Chronic Pelvic Pain Syndrome Evaluated by fMRI"

_brainsci, 2022, doi:10.3390/brainsci12101344_

Round 1

Reviewer 1 Report

This is an interesting fMRI study on the brain functional alterations of spontaneous pelvic pain in the patients of chronic pelvic pain syndrome. I have several major concerns in this study:

1. There is a need for the authors to clarify further how the chronic prostatitis was diagnosed in the current study.

2. The sample size of the current study is notably small for a proper between-subject comparison. I am concerned with the credibility of the finding. Power analysis is necessary to ensure that the result is not an artifact of small sample size.

Button, K. S., Ioannidis, J., Mokrysz, C., Nosek, B. A., Flint, J., Robinson, E. S., & Munafò, M. R. (2013). Power failure: why small sample size undermines the reliability of neuroscience. Nature reviews neuroscience, 14(5), 365-376.   3. More information should be provided regarding the comorbidity of the diseases in CP/CPPS patients   4. To increase the credibility of the finding, I am strongly recommend the authors to provide their data, codes, and materials in public database.   5. Minor comment: I noticed many grammatical errors and unclear sentences in the manuscript. It will be important for the authors to carefully proofread their manuscript further.

Author Response

Response to Reviewer 1 Comments

1. There is a need for the authors to clarify further how the chronic prostatitis was diagnosed in the current study.

Response 1: As a pilot study with a little cohort, we discovered the dysfunction of several brain regions might be related to the pain or emotional symptoms in CP/CPPS through this work. This provided us confidence to design following research. We are recruiting a larger cohort with alternative treatment and placebo treatment to investigate the potential biomarkers for diagnosis and prognosis.

2. The sample size of the current study is notably small for a proper between-subject comparison. I am concerned with the credibility of the finding. Power analysis is necessary to ensure that the result is not an artifact of small sample size.

Response 2: In fact, we utilized the False Discovery Rates (FDR), as multiple comparison correction, to correct the values of abnormal brain regions (voxel p<0.01, cluster p<0.01).

3. More information should be provided regarding the comorbidity of the diseases in CP/CPPS patients.

Response 3:

We added following statements:

Chief complaint, physical examinations, routine urine, standard microbiological cul-tures of urine, and transrectal ultrasonography (TRUS) were used for the diagnosis of CP/CPPS and the exclusion of acute or chronic bacterial prostatitis, benign prostate hy-perplasia, prostate cancer, and other pelvic diseases. The participants were excluded if they admitted to get any other chronic pain disorders, history of malignant tumors, and chronic diseases that might lead to peripheral nerve injury like diabetes mellitus and hypertension.

4. To increase the credibility of the finding, I am strongly recommend the authors to provide their data, codes, and materials in public database.

Response 4: Although there was a small sample data, there were also many indexes of fMRI for further analysis. If this manuscript were accepted, you could contact with the corresponding authors for asking the raw data.

5. Minor comment: I noticed many grammatical errors and unclear sentences in the manuscript. It will be important for the authors to carefully proofread their manuscript further.

Response 5:

We revised the manuscript carefully, and we would follow your kind advice in heart.

Reviewer 2 Report

The brain functional alterations of spontaneous pelvic pain in the patients of chronic prostatitis/chronic pelvic pain syndrome evaluated by fMRI
(brainsci-1908833)
(Review)

Main message of the article

The article “The brain functional alterations of spontaneous pelvic pain in the patients of chronic prostatitis/chronic pelvic pain syndrome evaluated by fMRI” by Ge and colleagues explores the functional brain activity in a group of 20 patients with chronic prostatitis/chronic pelvic pain syndrome (CP/CPPS) and 20 healthy participants. Results showed abnormal activity in the thalamus, in the inferior parietal lobule, and cingulate gyrus in patients with CP/CPPS. Furthermore, correlations between brain activity, pain intensity and HADS anxiety subscale were observed.

General Judgment Comments

The study is interesting, and the analysis are well conducted. While the title provides the needed information, the keywords used to index the article are not very informative in their current state and I would suggest editing them. The abstract is clear, but it needs some information to frame the sample and the results appropriately. Figures need some more work as the brain activation is not clearly visible and the caption are not self- explanatory. Further information should be provided to clarify how the participants were recruited and how the optimal sample size was determined. All results need to be reported in the standard format and effect size for significant results should be computed.

I would recommend the manuscript to undergo Major Revision.

Major Issues

  • -  Please change the keywords to make them informative for the reader and to increase the visibility of the study.

  • -  In Figure 1, results in terms of brain activation are not clearly visible. Furthermore, the caption should be self-explanatory and all abbreviations (i.e., mALFF, FDR) should be clarified by providing their extended version. Please modify accordingly also Figure 2 and 3.

  • -  Please provide the main demographic information (i.e., age and gender distribution) about the sample in the abstract.

  • -  In the Abstract, please report results in the standard format.

  • -  From the abstract, it is not clear what HADS refers to.

  • -  “Therefore, we delineated the functional alterations in brain might attribute to the abnormality of thalamus, inferior parietal lobule and

    cingulate gyrus”. What do the authors mean with this sentence? Please clarify.

  • -  From the Introduction, the aims of the current study are not clear. Please clarify.

  • -  Abbreviations in Tables should be explained in the caption.

  • -  How was the sample size determined?

  • -  How were participants recruited?

  • -  Please specify the ethical aspects of the study in the main text.

  • -  Please provide further details for the adopted scales.

  • -  At Lines 81-82, the authors stated that 20 healthy participants were recruited as control group while in the Abstract they say the control

    group consisted of 30 participants. Please clarify.

  • -  Was any quality check conducted on the obtained fMRI images? How did the authors asses the quality of the pre-processing and the

    images?

  • -  In the main text of the manuscript, results are often not reported in the standard format. Please modify.

  • -  What is the effect size for the significant results?

    Minor Issues

  • -  Lines 70-71: “Our previous work revealed that altered functional connectivities in the patients of CP/CPPS by analyzing the default mode network (DMN) in the resting-state, which was associated with the clinic pain intensity and anxiety condition”. This sentence is unclear. Please clarify.

  • -  Line 74: “central nerve system” should be “central nervous system”.

Final comments

I would recommend the manuscript to undergo Major Revision.

Author Response

Major Issues

  1. Please change the keywords to make them informative for the reader and to increase the visibility of the study.

Response 1: We corrected the abbreviation into related informative full names.

  1. In Figure 1, results in terms of brain activation are not clearly visible. Furthermore, the caption should be self-explanatory and all abbreviations (i.e., mALFF, FDR) should be clarified by providing their extended version. Please modify accordingly also Figure 2 and

Response 2: We modified the responding sites.

  1. Please provide the main demographic information (i.e., age and gender distribution) about the sample in the abstract.

Response 3: We provided related information.

  1. In the Abstract, please report results in the standard format.

Response 4: We corrected this point.

  1. From the abstract, it is not clear what HADS refers to.

Response 5: We clarified this aberration.

  1. “Therefore, we delineated the functional alterations in brain might attribute to the abnormality of thalamus, inferior parietal lobule and cingulate gyrus”. What do the authors mean with this sentence? Please clarify.

Response 6: This sentence should be “Therefore, we delineated the clinical alterations in patients of CP/CPPS might attribute to the functional abnormality of thalamus, inferior parietal lobule and cingulate gyrus.”

  1. From the Introduction, the aims of the current study are not clear. Please clarify.

Response 7: “to discovery the possible functional alterations of brain in the resting state and elucidate the potential mechanism of the pathogenesis and development in CP/CPPS”, which we further wrote in the manuscript.

  1. Abbreviations in Tables should be explained in the caption.

Response 8: We added the supplement in the caption.

  1. How was the sample size determined?

Response 9: According to the previous study, 15 participants in each group are demanded for the minimum size in neuroimaging studies (1). Then, we calculated the sample size according to the remaining research funds.

  1. How were participants recruited?

Response 10: The patients of CP/CPPS were straightly recruited by oral inquiry and fully informed of the project in the daily clinic of Huashan Hospital, Shanghai, China whilst the group of healthy control was recruited by poster advertising, which was added in the chapter Characteristics of participants in red font.

  1. Please specify the ethical aspects of the study in the main text.

Response 11: The ethical statement was added in the chapter Characteristics of participants in red font.

  1. Please provide further details for the adopted scales.

Response 12: We made some supplement to introduce the adapted clinical scales in the chapter Characteristics of participants in red font.

  1. At Lines 81-82, the authors stated that 20 healthy participants were recruited as control group while in the Abstract they say the control group consisted of 30 participants. Please clarify.

Response 13: We corrected this mistake.

  1. Was any quality check conducted on the obtained fMRI images? How did the authors asses the quality of the pre-processing and the images?

Response 14: We did check the fMRI images by a senior neuroradiologist Dr. Zhang in the early process of fMRI data acquisition. “To avoid statistic errors, all the scans were well examined before preprocessed. Unfortunately, 2 sets of data in patient group were abandoned because of image distortion.”, which was mentioned in the chapter Data preprocessing analysis.

  1. In the main text of the manuscript, results are often not reported in the standard format. Please modify.

 Response 15: We modified this point.

  1. What is the effect size for the significant results?

Response 16: In the correlation analysis, the R2 were 0.4620, 0.2261 and 0.2667, respectively.

Minor Issues

  1. Lines 70-71: “Our previous work revealed that altered functional connectivities in the patients of CP/CPPS by analyzing the default mode network (DMN) in the resting-state, which was associated with the clinic pain intensity and anxiety condition”. This sentence is unclear. Please clarify.

Response 17: This sentence should be “Our previous work revealed that altered functional connectivities in the patients of CP/CPPS might play a vital role in the pathogenesis and development of CP/CPPS by analyzing the integrity of default mode network (DMN)”, which we corrected in red font.

  1. Line 74: “central nerve system” should be “central nervous system”.

Response 17: We corrected this mistake.

reference

  1. Hayasaka S, Peiffer AM, Hugenschmidt CE, Laurienti PJ. Power and sample size calculation for neuroimaging studies by non-central random field theory. Neuroimage. 2007;37(3):721-30. Epub 2007/07/31. doi: 10.1016/j.neuroimage.2007.06.009. PubMed PMID: 17658273; PMCID: PMC2041809.

Reviewer 3 Report

The authors analyzed mALFF and mfALFF of fMRI in CP/CPPS patients and healthy volunteers. They found that CP/CPPS group has significantly higher mALFF values in the thalamus and augmented fALFF values in the inferior parietal lobule and cingulate gyrus. However, the brain targets the authors studies are regions relevant to general stress and pain. Therefore, the authors need to provide strong evidence of how these regions could be biomarkers for the diagnosis and potential treatment of CP/CPPS. Otherwise, the manuscript might lack scientific significance.

Line 184-185: “we mainly in the dorsomedial Moreover, the”. Please fix this typo.

Figure 2, please add scalar bar in the MRI images. Also, the pixels overlapped with the MRI images are very small. It looks like all the pixels are red. Is it possible to give a few zoomed in examples?

Figure 3, please add color bars and scalar bars.

It looks like Figure 3a only has 16 data points rather than 18? Why is that?

Line 240-246, perhaps the definitions of ALFF and fALFF could be discussed in Methods instead of in Discussion.

Line 316, what does ‘by lucubrate’ mean?

For the data analysis, should the authors perform multiple comparison correction since multiple ROIs were compared? If not, please justify.

Author Response

Response to Reviewer 3 Comments

The authors analyzed mALFF and mfALFF of fMRI in CP/CPPS patients and healthy volunteers. They found that CP/CPPS group has significantly higher mALFF values in the thalamus and augmented fALFF values in the inferior parietal lobule and cingulate gyrus. However, the brain targets the authors studies are regions relevant to general stress and pain. Therefore, the authors need to provide strong evidence of how these regions could be biomarkers for the diagnosis and potential treatment of CP/CPPS. Otherwise, the manuscript might lack scientific significance.

Response: As a pilot study with a little cohort, we discovered the dysfunction of several brain regions might be related to the pain or emotional symptoms in CP/CPPS through this work. This provided us confidence to design following research. We are recruiting a larger cohort with alternative treatment and placebo treatment to investigate the potential biomarkers for diagnosis and prognosis.

  1. Line 184-185: “we mainly in the dorsomedial Moreover, the”. Please fix this typo.

Response 1: We corrected this mistake.

  1. Figure 2, please add scalar bar in the MRI images. Also, the pixels overlapped with the MRI images are very small. It looks like all the pixels are red. Is it possible to give a few zoomed in examples?

Response 2: We followed your advice to improve.

  1. Figure 3, please add color bars and scalar bars.

Response 3: We added the color bars. As we used the calibrated standard brain map with pseudo-color, it’s difficult to add scalar bars.

  1. It looks like Figure 3a only has 16 data points rather than 18? Why is that?

Response 4: Because there were very closed values in two group of data (score 2 and 5). Therefore, two points of these groups almost coincided.

  1. Line 240-246, perhaps the definitions of ALFF and fALFF could be discussed in Methods instead of in Discussion.

Response 5: We moved the definitions of ALFF and fALFF into the Introduction.

  1. Line 316, what does ‘by lucubrate’ mean?

Response 6: We corrected this mistake in red font.

  1. For the data analysis, should the authors perform multiple comparison correction since multiple ROIs were compared? If not, please justify.

Response 7: We did utilize the multiple comparison correction of “False Discovery Rates (FDR), as multiple comparison correction, corrected the values of abnormal brain regions (voxel p<0.01, cluster p<0.01)” stated in the Methods - Statistics Analysis.

Round 2

Reviewer 1 Report

The authors have sufficiently addressed my comments.

Author Response

Thank you for your kind advice.

Reviewer 2 Report

The ALFF alterations of spontaneous pelvic pain in the patients of chronic prostatitis/chronic pelvic pain syndrome evaluated by fMRI

(brainsci-1908833-peer-review-v2)
(Review)

The manuscript has been improved and I thank the authors for their work on the revision. Some minor amendments are required.

I would recommend the manuscript to undergo Minor Revision.

Minor Issues:

  • -  The authors stated that the provided the main demographic information in the abstract (see point 3 of the response letter). However, I am still not able to see the edited part. Please specify or add the requested information to the abstract.

  • -  The authors stated “Response 9: According to the previous study, 15 participants in each group are demanded for the minimum size in neuroimaging studies (1). Then, we calculated the sample size according to the remaining research funds”. Please also specify this part in the text.

  • -  The authors reported that “Response 16: In the correlation analysis, the R2 were 0.4620, 0.2261 and 0.2667, respectively”. Please specify in the text too.

    I would recommend the manuscript to undergo Minor Revision.

Author Response

-  The authors stated that the provided the main demographic information in the abstract (see point 3 of the response letter). However, I am still not able to see the edited part. Please specify or add the requested information to the abstract.

Response 1: We added this point.

-  The authors stated “Response 9: According to the previous study, 15 participants in each group are demanded for the minimum size in neuroimaging studies (1). Then, we calculated the sample size according to the remaining research funds”. Please also specify this part in the text.

Response 2: We supplemented this part.

-  The authors reported that “Response 16: In the correlation analysis, the R2 were 0.4620, 0.2261 and 0.2667, respectively”. Please specify in the text too.

Response 3: We complemented this text.

Reviewer 3 Report

The authors have fully resolved my concerns. Thanks!

Author Response

Thank you for your kind advice!